# Identifying and attacking the saddle point problem in high-dimensional non-convex optimization

**Yann N. Dauphin  Razvan Pascanu  Caglar Gulcehre  Kyunghyun Cho**
Université de Montréal
dauphiya@iro.umontreal.ca, r.pascanu@gmail.com,
gulcehrc@iro.umontreal.ca, kyunghyun.cho@umontreal.ca

**Surya Ganguli**
Stanford University
sganguli@standford.edu

**Yoshua Bengio**
Université de Montréal, CIFAR Fellow
yoshua.bengio@umontreal.ca

## Abstract

A central challenge to many fields of science and engineering involves minimizing non-convex error functions over continuous, high dimensional spaces. Gradient descent or quasi-Newton methods are almost ubiquitously used to perform such minimizations, and it is often thought that a main source of difficulty for these local methods to find the global minimum is the proliferation of local minima with much higher error than the global minimum. Here we argue, based on results from statistical physics, random matrix theory, neural network theory, and empirical evidence, that a deeper and more profound difficulty originates from the proliferation of saddle points, not local minima, especially in high dimensional problems of practical interest. Such saddle points are surrounded by high error plateaus that can dramatically slow down learning, and give the illusory impression of the existence of a local minimum. Motivated by these arguments, we propose a new approach to second-order optimization, the saddle-free Newton method, that can rapidly escape high dimensional saddle points, unlike gradient descent and quasi-Newton methods. We apply this algorithm to deep or recurrent neural network training, and provide numerical evidence for its superior optimization performance.

## 1 Introduction

It is often the case that our geometric intuition, derived from experience within a low dimensional physical world, is inadequate for thinking about the geometry of typical error surfaces in high-dimensional spaces. To illustrate this, consider minimizing a randomly chosen error function of a single scalar variable, given by a single draw of a Gaussian process. (Rasmussen and Williams, 2005) have shown that such a random error function would have many local minima and maxima, with high probability over the choice of the function, but saddles would occur with negligible probability. On the other-hand, as we review below, typical, random Gaussian error functions over $N$ scalar variables, or dimensions, are increasingly likely to have saddle points rather than local minima as $N$ increases. Indeed the ratio of the number of saddle points to local minima increases *exponentially* with the dimensionality $N$.

A typical problem for both local minima and saddle-points is that they are often surrounded by plateaus of small curvature in the error. While gradient descent dynamics are repelled away from a saddle point to lower error by following directions of negative curvature, this repulsion can occur slowly due to the plateau. Second order methods, like the Newton method, are designed to rapidly descend plateaus surrounding local minima by multiplying the gradient steps with the inverse of the Hessian matrix. However, the Newton method does not treat saddle points appropriately; as argued below, saddle-points instead become *attractive* under the Newton dynamics.

Thus, given the proliferation of saddle points, not local minima, in high dimensional problems, the entire theoretical justification for quasi-Newton methods, i.e. the ability to rapidly descend to the bottom of a convex local minimum, becomes less relevant in high dimensional non-convex optimization. In this work, which

is an extension of the previous report Pascanu *et al.* (2014), we first want to raise awareness of this issue, and second, propose an alternative approach to second-order optimization that aims to rapidly escape from saddle points. This algorithm leverages second-order curvature information in a fundamentally different way than quasi-Newton methods, and also, in numerical experiments, outperforms them in some high dimensional problems involving deep or recurrent networks.

## 2 The prevalence of saddle points in high dimensions

Here we review arguments from disparate literatures suggesting that saddle points, not local minima, provide a fundamental impediment to rapid high dimensional non-convex optimization. One line of evidence comes from statistical physics. Bray and Dean (2007); Fyodorov and Williams (2007) study the nature of critical points of random Gaussian error functions on high dimensional continuous domains using replica theory (see Parisi (2007) for a recent review of this approach).

One particular result by Bray and Dean (2007) derives how critical points are distributed in the $\epsilon$ vs $\alpha$ plane, where $\alpha$ is the index, or the fraction of negative eigenvalues of the Hessian at the critical point, and $\epsilon$ is the error attained at the critical point. Within this plane, critical points concentrate on a monotonically increasing curve as $\alpha$ ranges from 0 to 1, implying a strong correlation between the error $\epsilon$ and the index $\alpha$: the larger the error the larger the index. The probability of a critical point to be an $O(1)$ distance off the curve is exponentially small in the dimensionality $N$, for large $N$. This implies that critical points with error $\epsilon$ much larger than that of the global minimum, are exponentially likely to be saddle points, with the fraction of negative curvature directions being an increasing function of the error. Conversely, all local minima, which necessarily have index 0, are likely to have an error very close to that of the global minimum. Intuitively, *in high dimensions, the chance that all the directions around a critical point lead upward (positive curvature) is exponentially small* w.r.t. the number of dimensions, unless the critical point is the global minimum or stands at an error level close to it, i.e., it is unlikely one can find a way to go further down.

These results may also be understood via random matrix theory. We know that for a large Gaussian random matrix the eigenvalue distribution follows Wigner's famous semicircular law (Wigner, 1958), with both mode and mean at 0. The probability of an eigenvalue to be positive or negative is thus $\frac{1}{2}$. Bray and Dean (2007) showed that the eigenvalues of the Hessian at a critical point are distributed in the same way, except that the semicircular spectrum is shifted by an amount determined by $\epsilon$. For the global minimum, the spectrum is shifted so far right, that all eigenvalues are positive. As $\epsilon$ increases, the spectrum shifts to the left and accrues more negative eigenvalues as well as a density of eigenvalues around 0, indicating the typical presence of plateaus surrounding saddle points at large error. Such plateaus would slow the convergence of first order optimization methods, yielding the illusion of a local minimum.

The random matrix perspective also concisely and intuitively crystallizes the striking difference between the geometry of low and high dimensional error surfaces. For $N = 1$, an exact saddle point is a 0–probability event as it means randomly picking an eigenvalue of exactly 0. As $N$ grows it becomes exponentially unlikely to randomly pick all eigenvalues to be positive or negative, and therefore most critical points are saddle points.

Fyodorov and Williams (2007) review qualitatively similar results derived for random error functions superimposed on a quadratic error surface. These works indicate that for typical, generic functions chosen from a random Gaussian ensemble of functions, local minima with high error are exponentially rare in the dimensionality of the problem, but saddle points with many negative and approximate plateau directions are exponentially likely. However, is this result for generic error landscapes applicable to the error landscapes of practical problems of interest?

Baldi and Hornik (1989) analyzed the error surface of a multilayer perceptron (MLP) with a single linear hidden layer. Such an error surface shows only saddle-points and *no* local minima. This result is qualitatively consistent with the observation made by Bray and Dean (2007). Indeed Saxe *et al.* (2014) analyzed the dynamics of learning in the presence of these saddle points, and showed that they arise due to scaling symmetries in the weight space of a deep linear MLP. These scaling symmetries enabled Saxe *et al.* (2014) to find new exact solutions to the nonlinear dynamics of learning in deep linear networks. These learning dynamics exhibit plateaus of high error followed by abrupt transitions to better performance. They qualitatively recapitulate aspects of the hierarchical development of semantic concepts in infants (Saxe *et al.*, 2013).

In (Saad and Solla, 1995) the dynamics of stochastic gradient descent are analyzed for soft committee machines. This work explores how well a student network can learn to imitate a randomly chosen teacher network. Importantly, it was observed that learning can go through an initial phase of *being trapped in the symmetric submanifold* of weight space. In this submanifold, the student's hidden units compute similar functions over the distribution of inputs. The slow learning dynamics within this submanifold originates from saddle point structures (caused by permutation symmetries among hidden units), and their associated

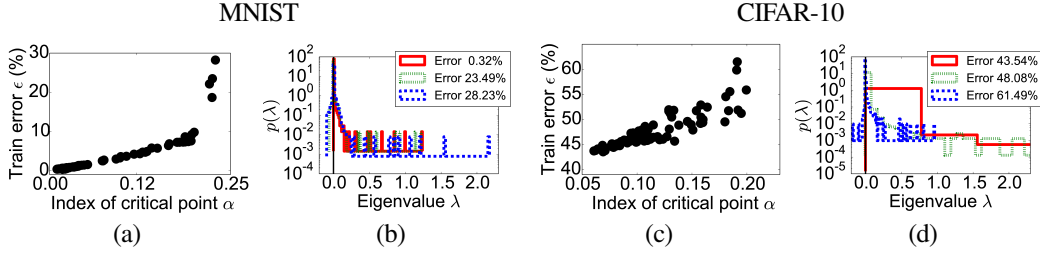

Figure 1: (a) and (c) show how critical points are distributed in the $\epsilon$–$\alpha$ plane. Note that they concentrate along a monotonically increasing curve. (b) and (d) plot the distributions of eigenvalues of the Hessian at three different critical points. Note that the y axes are in logarithmic scale. The vertical lines in (b) and (d) depict the position of 0.

plateaus (Rattray *et al.*, 1998; Inoue *et al.*, 2003). The exit from the plateau associated with the symmetric submanifold corresponds to the differentiation of the student's hidden units to mimic the teacher's hidden units. Interestingly, this exit from the plateau is achieved by following directions of negative curvature associated with a saddle point. sin directions perpendicular to the symmetric submanifold.

Mizutani and Dreyfus (2010) look at the effect of negative curvature on learning and implicitly at the effect of saddle points in the error surface. Their findings are similar. They show that the error surface of a single layer MLP has saddle points where the Hessian matrix is indefinite.

# 3 Experimental validation of the prevalence of saddle points

In this section, we experimentally test whether the theoretical predictions presented by Bray and Dean (2007) for random Gaussian fields hold for neural networks. To our knowledge, this is the first attempt to measure the relevant statistical properties of neural network error surfaces and to test if the theory developed for random Gaussian fields generalizes to such cases.

In particular, we are interested in how the critical points of a single layer MLP are distributed in the $\epsilon$–$\alpha$ plane, and how the eigenvalues of the Hessian matrix at these critical points are distributed. We used a small MLP trained on a down-sampled version of MNIST and CIFAR-10. Newton method was used to identify critical points of the error function. The results are in Fig. 1. More details about the setup are provided in the supplementary material.

This empirical test confirms that the observations by Bray and Dean (2007) qualitatively hold for neural networks. Critical points concentrate along a monotonically increasing curve in the $\epsilon$–$\alpha$ plane. Thus the prevalence of high error saddle points do indeed pose a severe problem for training neural networks. While the eigenvalues do not seem to be exactly distributed according to the semicircular law, their distribution does shift to the left as the error increases. The large mode at 0 indicates that there is a plateau around any critical point of the error function of a neural network.

# 4 Dynamics of optimization algorithms near saddle points

Given the prevalence of saddle points, it is important to understand how various optimization algorithms behave near them. Let us focus on non-degenerate saddle points for which the Hessian is not singular. These critical points can be locally analyzed by re-parameterizing the function according to Morse's lemma below (see chapter 7.3, Theorem 7.16 in Callahan (2010) or the supplementary material for details):

$$f(\theta^* + \Delta\theta) = f(\theta^*) + \frac{1}{2}\sum_{i=1}^{n_\theta} \lambda_i \Delta\mathbf{v}_i^2, \tag{1}$$

where $\lambda_i$ represents the $i$th eigenvalue of the Hessian, and $\Delta\mathbf{v}_i$ are the new parameters of the model corresponding to motion along the eigenvectors $\mathbf{e}_i$ of the Hessian of $f$ at $\theta^*$.

If finding the local minima of our function is the desired outcome of our optimization algorithm, we argue that an optimal algorithm would move away from the saddle point at a speed that is inverse proportional with the flatness of the error surface and hence depndented of how trustworthy this descent direction is further away from the current position.

A step of the *gradient descent* method always points away from the saddle point close to it (SGD in Fig. 2). Assuming equation (1) is a good approximation of our function we will analyze the optimality of the step according to how well the resulting $\Delta\mathbf{v}$ optimizes the right hand side of (1). If an eigenvalue $\lambda_i$ is positive (negative), then the step moves toward (away) from $\theta^*$ along $\Delta\mathbf{v}_i$ because the restriction of $f$ to the corresponding eigenvector direction $\Delta\mathbf{v}_i$, achieves a minimum (maximum) at $\theta^*$. The drawback of the gradient descent method is not the direction, but the *size* of the step along each eigenvector. The step, along any direction $\mathbf{e}_i$, is given by $-\lambda_i\Delta\mathbf{v}_i$, and so small steps are taken in directions corresponding to eigenvalues of small absolute value.

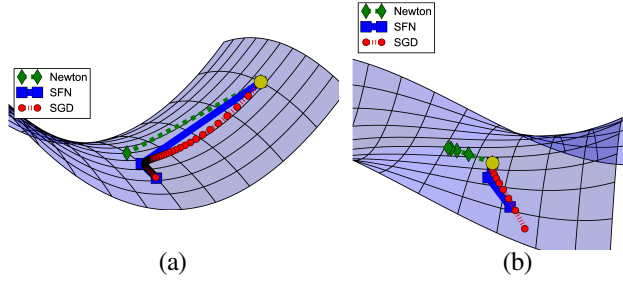

(a)  (b)

Figure 2: Behaviors of different optimization methods near a saddle point for (a) classical saddle structure $5x^2 - y^2$; (b) monkey saddle structure $x^3 - 3xy^2$. The yellow dot indicates the starting point. SFN stands for the saddle-free Newton method we proposed.

The *Newton method* solves the slowness problem by rescaling the gradients in each direction with the inverse of the corresponding eigenvalue, yielding the step $-\Delta\mathbf{v}_i$. However, this approach can result in moving toward the saddle point. Specifically, if an eigenvalue is negative, the Newton step moves along the eigenvector in a direction *opposite* to the gradient descent step, and thus moves in the direction of $\theta^*$. $\theta^*$ becomes an *attractor* for the Newton method (see Fig. 2), which can get stuck in this saddle point and not converge to a local minima. This justifies using the Newton method to find critical points of any index in Fig. 1.

A *trust region* approach is one approach of scaling second order methods to non-convex problems. In one such method, the Hessian is damped to remove negative curvature by adding a constant $\alpha$ to its diagonal, which is equivalent to adding $\alpha$ to each of its eigenvalues. If we project the new step along the different eigenvectors of the modified Hessian, it is equivalent to rescaling the projections of the gradient on this direction by the inverse of the modified eigenvalues $\lambda_i + \alpha$ yields the step $-\left(\lambda_i/\lambda_i+\alpha\right)\Delta\mathbf{v}_i$. To ensure the algorithm does not converge to the saddle point, one must increase the damping coefficient $\alpha$ enough so that $\lambda_{min}+\alpha > 0$ even for the most negative eigenvalue $\lambda_{min}$. This ensures that the modified Hessian is positive definnite. However, the drawback is again a potentially small step size in many eigen-directions incurred by a large damping factor $\alpha$ (the rescaling factors in each eigen-direction are not proportional to the curvature anymore).

Besides damping, another approach to deal with negative curvature is to ignore them. This can be done regardless of the approximation strategy used for the Newton method such as a truncated Newton method or a BFGS approximation (see Nocedal and Wright (2006) chapters 4 and 7). However, such algorithms cannot escape saddle points, as they ignore the very directions of negative curvature that must be followed to achieve escape.

Natural gradient descent is a first order method that relies on the curvature of the parameter manifold. That is, natural gradient descent takes a step that induces a constant change in the behaviour of the model as measured by the KL-divergence between the model before and after taking the step. The resulting algorithm is similar to the Newton method, except that it relies on the Fisher Information matrix $\mathbf{F}$.

It is argued by Rattray *et al.* (1998); Inoue *et al.* (2003) that natural gradient descent can address certain saddle point structures effectively. Specifically, it can resolve those saddle points arising from having units behaving very similarly. Mizutani and Dreyfus (2010), however, argue that natural gradient descent also suffers with negative curvature. One particular known issue is the over-realizable regime, where around the stationary solution $\theta^*$, the Fisher matrix is rank-deficient. Numerically, this means that the Gauss-Newton direction can be orthogonal to the gradient at some distant point from $\theta^*$ (Mizutani and Dreyfus, 2010), causing optimization to converge to some non-stationary point. Another weakness is that the difference $\mathbf{S}$ between the Hessian and the Fisher Information Matrix can be large near certain saddle points that exhibit strong negative curvature. This means that the landscape close to these critical points may be dominated by $\mathbf{S}$, meaning that the rescaling provided by $\mathbf{F}^{-1}$ is not optimal in all directions.

The same is true for TONGA (Le Roux *et al.*, 2007), an algorithm similar to natural gradient descent. It uses the covariance of the gradients as the rescaling factor. As these gradients vanish approaching a critical point, their covariance will result in much larger steps than needed near critical points.

# 5 Generalized trust region methods

In order to attack the saddle point problem, and overcome the deficiencies of the above methods, we will define a class of *generalized trust region methods*, and search for an algorithm within this space. This class involves a straightforward extension of classical trust region methods via two simple changes: (1) We allow the minimization of a first-order Taylor expansion of the function instead of always relying on a second-order Taylor expansion as is typically done in trust region methods, and (2) we replace the constraint on the norm of the step $\Delta\theta$ by a constraint on the distance between $\theta$ and $\theta+\Delta\theta$. Thus the choice of distance function and Taylor expansion order specifies an algorithm. If we define $\mathcal{T}_k(f,\theta,\Delta\theta)$ to indicate the $k$-th order Taylor series expansion of $f$ around $\theta$ evaluated at $\theta+\Delta\theta$, then we can summarize a generalized trust region method as:

$$\Delta\theta = \operatorname*{argmin}_{\Delta\theta} \mathcal{T}_k\{f,\theta,\Delta\theta\} \quad \text{with } k \in \{1,2\} \text{ s. t. } d(\theta,\theta+\Delta\theta) \leq \Delta. \tag{2}$$

For example, the $\alpha$-damped Newton method described above arises as a special case with $k=2$ and $d(\theta,\theta+\Delta\theta)=||\Delta\theta||_2^2$, where $\alpha$ is implicitly a function of $\Delta$.

# 6 Attacking the saddle point problem

---

**Algorithm 1** Approximate saddle-free Newton

---

**Require:** Function $f(\theta)$ to minimize
  **for** $i=1 \to M$ **do**
    $\mathbf{V} \leftarrow k$ Lanczos vectors of $\frac{\partial^2 f}{\partial\theta^2}$
    $s(\alpha) = f(\theta+\mathbf{V}\alpha)$
    $|\hat{\mathbf{H}}| \leftarrow \left|\frac{\partial^2 s}{\partial\alpha^2}\right|$ by using an eigen decomposition of $\hat{\mathbf{H}}$
    **for** $j=1 \to m$ **do**
      $\mathbf{g} \leftarrow -\frac{\partial s}{\partial\alpha}$
      $\lambda \leftarrow \operatorname{argmin}_\lambda s((|\hat{\mathbf{H}}|+\lambda\mathbf{I})^{-1}\mathbf{g})$
      $\theta \leftarrow \theta + \mathbf{V}(|\hat{\mathbf{H}}|+\lambda\mathbf{I})^{-1}\mathbf{g}$
    **end for**
  **end for**

---

We now search for a solution to the saddle-point problem within the family of generalized trust region methods. In particular, the analysis of optimization algorithms near saddle points discussed in Sec. 4 suggests a simple heuristic solution: rescale the gradient along each eigen-direction $\mathbf{e}_i$ by $1/|\lambda_i|$. This achieves the same optimal rescaling as the Newton method, while preserving the sign of the gradient, thereby turning saddle points into repellers, not attractors, of the learning dynamics. The idea of taking the absolute value of the eigenvalues of the Hessian was suggested before. See, for example, (Nocedal and Wright, 2006, chapter 3.4) or Murray (2010, chapter 4.1). However, we are not aware of any proper justification of this algorithm or even a detailed exploration (empirical or otherwise) of this idea. One cannot simply replace $\mathbf{H}$ by $|\mathbf{H}|$, where $|\mathbf{H}|$ is the matrix obtained by taking the absolute value of each eigenvalue of $\mathbf{H}$, without proper justi-fication. While we might be able to argue that this heuristic modification does the right thing near critical points, is it still the right thing far away from the critical points? How can we express this step in terms of the existing methods ? Here we show this heuristic solution arises naturally from our generalized trust region approach.

Unlike classical trust region approaches, we consider minimizing a first-order Taylor expansion of the loss ($k=1$ in Eq. (2)). This means that the curvature information has to come from the constraint by picking a suitable distance measure $d$ (see Eq. (2)). Since the minimum of the first order approximation of $f$ is at infinity, we know that this optimization dynamics will always jump to the border of the trust region. So we must ask how far from $\theta$ can we trust the first order approximation of $f$? One answer is to bound the discrepancy between the first and second order Taylor expansions of $f$ by imposing the following constraint:

$$d(\theta,\theta+\Delta\theta) = \left| f(\theta) + \nabla f \Delta\theta + \frac{1}{2}\Delta\theta^\top \mathbf{H}\Delta\theta - f(\theta) - \nabla f \Delta\theta \right| = \frac{1}{2}\left| \Delta\theta^\top \mathbf{H}\Delta\theta \right| \leq \Delta, \tag{3}$$

where $\nabla f$ is the partial derivative of $f$ with respect to $\theta$ and $\Delta \in \mathbb{R}$ is some small value that indicates how much discrepancy we are willing to accept. Note that the distance measure $d$ takes into account the curvature of the function.

Eq. (3) is not easy to solve for $\Delta\theta$ in more than one dimension. Alternatively, one could take the square of the distance, but this would yield an optimization problem with a constraint that is quartic in $\Delta\theta$, and therefore also difficult to solve. We circumvent these difficulties through a Lemma:

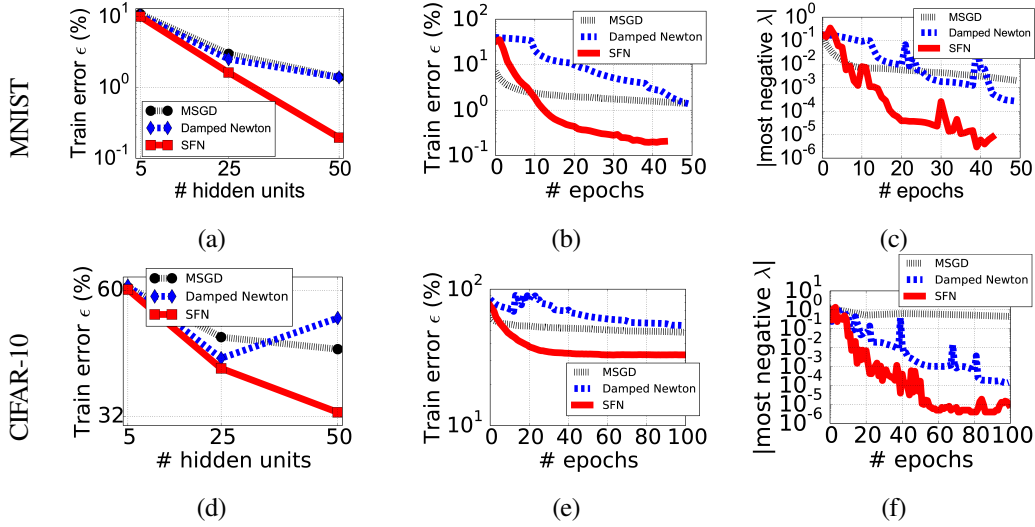

Figure 3: Empirical evaluation of different optimization algorithms for a single-layer MLP trained on the rescaled MNIST and CIFAR-10 dataset. In (a) and (d) we look at the minimum error obtained by the different algorithms considered as a function of the model size. (b) and (e) show the optimal training curves for the three algorithms. The error is plotted as a function of the number of epochs. (c) and (f) track the norm of the largest negative eigenvalue.

**Lemma 1.** *Let* $\mathbf{A}$ *be a nonsingular square matrix in* $\mathbb{R}^n \times \mathbb{R}^n$*, and* $\mathbf{x} \in \mathbb{R}^n$ *be some vector. Then it holds that* $|\mathbf{x}^\top \mathbf{A} \mathbf{x}| \leq \mathbf{x}^\top |\mathbf{A}| \mathbf{x}$*, where* $|\mathbf{A}|$ *is the matrix obtained by taking the absolute value of each of the eigenvalues of* $\mathbf{A}$*.*

*Proof.* See the supplementary material for the proof. □

Instead of the originally proposed distance measure in Eq. (3), we approximate the distance by its upper bound $\Delta\theta |\mathbf{H}| \Delta\theta$ based on Lemma 1. This results in the following generalized trust region method:

$$\Delta\theta = \underset{\Delta\theta}{\arg\min} f(\theta) + \nabla f \Delta\theta \quad \text{s. t. } \Delta\theta^\top |\mathbf{H}| \Delta\theta \leq \Delta. \tag{4}$$

Note that as discussed before, we can replace the inequality constraint with an equality one, as the first order approximation of $f$ has a minimum at infinity and the algorithm always jumps to the border of the trust region. Similar to (Pascanu and Bengio, 2014), we use Lagrange multipliers to obtain the solution of this constrained optimization. This gives (up to a scalar that we fold into the learning rate) a step of the form:

$$\Delta\theta = -\nabla f |\mathbf{H}|^{-1} \tag{5}$$

This algorithm, which we call the saddle-free Newton method (SFN), leverages curvature information in a fundamentally different way, to define the shape of the trust region, rather than Taylor expansion to second order, as in classical methods. Unlike gradient descent, it can move further (less) in the directions of low (high) curvature. It is identical to the Newton method when the Hessian is positive definite, but unlike the Newton method, it can escape saddle points. Furthermore, unlike gradient descent, the escape is rapid even along directions of weak negative curvature (see Fig. 2).

The exact implementation of this algorithm is intractable in a high dimensional problem, because it requires the exact computation of the Hessian. Instead we use an approach similar to Krylov subspace descent (Vinyals and Povey, 2012). We optimize that function in a lower-dimensional Krylov subspace $\hat{f}(\alpha) = f(\theta + \alpha \mathbf{V})$. The $k$ Krylov subspace vectors $\mathbf{V}$ are found through Lanczos iteration of the Hessian. These vectors will span the $k$ biggest eigenvectors of the Hessian with high-probability. This reparametrization through $\alpha$ greatly reduces the dimensionality and allows us to use exact saddle-free Newton in the subspace.[1] See Alg. 1 for the pseudocode.

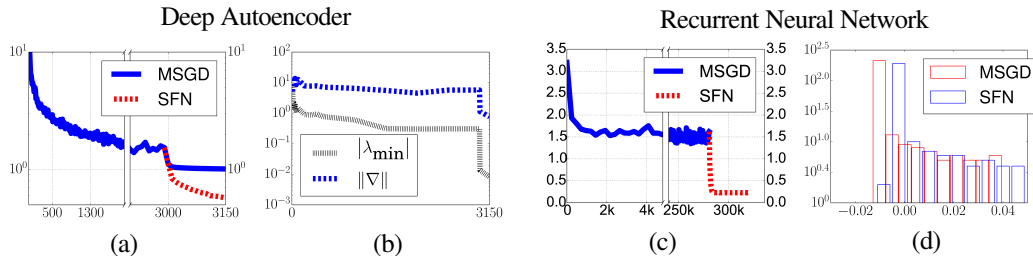

Figure 4: Empirical results on training deep autoencoders on MNIST and recurrent neural network on Penn Treebank. (a) and (c): The learning curve for SGD and SGD followed by saddle-free Newton method. (b) The evolution of the magnitude of the most negative eigenvalue and the norm of the gradients w.r.t. the number of epochs (deep autoencoder). (d) The distribution of eigenvalues of the RNN solutions found by SGD and the SGD continued with saddle-free Newton method.

## 7 Experimental validation of the saddle-free Newton method

In this section, we empirically evaluate the theory suggesting the existence of many saddle points in high-dimensional functions by training neural networks.

### 7.1 Existence of Saddle Points in Neural Networks

In this section, we validate the existence of saddle points in the cost function of neural networks, and see how each of the algorithms we described earlier behaves near them. In order to minimize the effect of any type of approximation used in the algorithms, we train small neural networks on the scaled-down version of MNIST and CIFAR-10, where we can compute the update directions by each algorithm exactly. Both MNIST and CIFAR-10 were downsampled to be of size $10 \times 10$.

We compare minibatch stochastic gradient descent (MSGD), damped Newton and the proposed saddle-free Newton method (SFN). The hyperparameters of SGD were selected via random search (Bergstra and Bengio, 2012), and the damping coefficients for the damped Newton and saddle-free Newton[2] methods were selected from a small set at each update.

The theory suggests that the number of saddle points increases exponentially as the dimensionality of the function increases. From this, we expect that it becomes more likely for the conventional algorithms such as SGD and Newton methods to stop near saddle points, resulting in worse performance (on training samples). Figs. 3 (a) and (d) clearly confirm this. With the smallest network, all the algorithms perform comparably, but as the size grows, the saddle-free Newton algorithm outperforms the others by a large margin.

A closer look into the different behavior of each algorithm is presented in Figs. 3 (b) and (e) which show the evolution of training error over optimization. We can see that the proposed saddle-free Newton escapes, or does not get stuck at all, near a saddle point where both SGD and Newton methods appear trapped. Especially, at the 10-th epoch in the case of MNIST, we can observe the saddle-free Newton method rapidly escaping from the saddle point. Furthermore, Figs. 3 (c) and (f) provide evidence that the distribution of eigenvalues shifts more toward the right as error decreases for all algorithms, consistent with the theory of random error functions. The distribution shifts more for SFN, suggesting it can successfully avoid saddle-points on intermediary error (and large index).

### 7.2 Effectiveness of saddle-free Newton Method in Deep Feedforward Neural Networks

Here, we further show the effectiveness of the proposed saddle-free Newton method in a larger neural network having seven hidden layers. The neural network is a deep autoencoder trained on (full-scale) MNIST and considered a standard benchmark problem for assessing the performance of optimization algorithms on neural networks (Sutskever *et al.*, 2013). In this large-scale problem, we used the Krylov subspace descent approach described earlier with 500 subspace vectors.

We first trained the model with SGD and observed that learning stalls after achieving the mean-squared error (MSE) of 1.0. We then continued with the saddle-free Newton method which rapidly escaped the (approximate) plateau at which SGD was stuck (See Fig. 4 (a)). Furthermore, even in these large scale

experiments, we were able to confirm that the distribution of Hessian eigenvalues shifts right as error decreases, and that the proposed saddle-free Newton algorithm accelerates this shift (See Fig. 4 (b)).

The model trained with SGD followed by the saddle-free Newton method was able to get the state-of-the-art MSE of 0.57 compared to the previous best error of 0.69 achieved by the Hessian-Free method (Martens, 2010). Saddle free Newton method does better.

### 7.3 Recurrent Neural Networks: Hard Optimization Problem

Recurrent neural networks are widely known to be more difficult to train than feedforward neural networks (see, e.g., Bengio *et al.*, 1994; Pascanu *et al.*, 2013). In practice they tend to underfit, and in this section, we want to test if the proposed saddle-free Newton method can help avoiding underfitting, assuming that that it is caused by saddle points. We trained a small recurrent neural network having 120 hidden units for the task of character-level language modeling on Penn Treebank corpus. Similarly to the previous experiment, we trained the model with SGD until it was clear that the learning stalled. From there on, training continued with the saddle-free Newton method.

In Fig. 4 (c), we see a trend similar to what we observed with the previous experiments using feedforward neural networks. The SGD stops progressing quickly and does not improve performance, suggesting that the algorithm is stuck in a plateau, possibly around a saddle point. As soon as we apply the proposed saddle-free Newton method, we see that the error drops significantly. Furthermore, Fig. 4 (d) clearly shows that the solution found by the saddle-free Newton has fewer negative eigenvalues, consistent with the theory of random Gaussian error functions. In addition to the saddle-free Newton method, we also tried continuing with the truncated Newton method with damping, however, without much success.

## 8   Conclusion

In summary, we have drawn from disparate literatures spanning statistical physics and random matrix theory to neural network theory, to argue that (a) non-convex error surfaces in high dimensional spaces generically suffer from a proliferation of saddle points, and (b) in contrast to conventional wisdom derived from low dimensional intuition, local minima with high error are exponentially rare in high dimensions. Moreover, we have provided the first experimental tests of these theories by performing new measurements of the statistical properties of critical points in neural network error surfaces. These tests were enabled by a novel application of Newton's method to search for critical points of any index (fraction of negative eigenvalues), and they confirmed the main qualitative prediction of theory that the index of a critical point tightly and positively correlates with its error level.

Motivated by this theory, we developed a framework of generalized trust region methods to search for algorithms that can rapidly escape saddle points. This framework allows us to leverage curvature information in a fundamentally different way than classical methods, by defining the shape of the trust region, rather than locally approximating the function to second order. Through further approximations, we derived an exceedingly simple algorithm, the saddle-free Newton method, which rescales gradients by the absolute value of the inverse Hessian. This algorithm had previously remained heuristic and theoretically unjustified, as well as numerically unexplored within the context of deep and recurrent neural networks. Our work shows that near saddle points it can achieve rapid escape by combining the best of gradient descent and Newton methods while avoiding the pitfalls of both. Moreover, through our generalized trust region approach, our work shows that this algorithm is sensible even far from saddle points. Finally, we demonstrate improved optimization on several neural network training problems.

For the future, we are mainly interested in two directions. The first direction is to explore methods beyond Kyrylov subspaces, such as one in (Sohl-Dickstein *et al.*, 2014), that allow the saddle-free Newton method to scale to high dimensional problems, where we cannot easily compute the entire Hessian matrix. In the second direction, the theoretical properties of critical points in the problem of training a neural network will be further analyzed. More generally, it is likely that a deeper understanding of the statistical properties of high dimensional error surfaces will guide the design of novel non-convex optimization algorithms that could impact many fields across science and engineering.

### Acknowledgments

We would like to thank the developers of Theano (Bergstra *et al.*, 2010; Bastien *et al.*, 2012). We would also like to thank CIFAR, and Canada Research Chairs for funding, and Compute Canada, and Calcul Québec for providing computational resources. Razvan Pascanu is supported by a DeepMind Google Fellowship. Surya Ganguli thanks the Burroughs Wellcome and Sloan Foundations for support.

## Footnotes

[1] In the Krylov subspace, $\frac{\partial \hat{f}}{\partial \alpha} = \mathbf{V} \left( \frac{\partial f}{\partial \theta} \right)^\top$ and $\frac{\partial^2 \hat{f}}{\partial \alpha^2} = \mathbf{V} \left( \frac{\partial^2 f}{\partial \theta^2} \right) \mathbf{V}^\top$.

[2]Damping is used for numerical stability.

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
