[Supplementary Material]

# Supplementary Material:
# Attacking the saddle point problem in
# high-dimensional non-convex optimization

**Yann N. Dauphin**
Université de Montréal
dauphiya@iro.umontreal.ca

**Razvan Pascanu**
Université de Montréal
r.pascanu@gmail.com

**Caglar Gulcehre**
Université de Montréal
gulcehrc@iro.umontreal.ca

**Kyunghyun Cho**
Université de Montréal
kyunghyun.cho@umontreal.ca

**Surya Ganguli**
Stanford University
sganguli@standford.edu

**Yoshua Bengio**
Université de Montréal
CIFAR Fellow
yoshua.bengio@umontreal.ca

## 1 Description of the different types of saddle-points

In general, consider an error function $f(\theta)$ where $\theta$ is an $N$ dimensional continuous variable. A critical point is by definition a point $\theta$ where the gradient of $f(\theta)$ vanishes. All critical points of $f(\theta)$ can be further characterized by the curvature of the function in its vicinity, as described by the eigenvalues of the Hessian. Note that the Hessian is symmetric and hence the eigenvalues are real numbers. The following are the four possible scenarios:

- If all eigenvalues are non-zero and positive, then the critical point is a local minimum.

- If all eigenvalues are non-zero and negative, then the critical point is a local maximum.

- If the eigenvalues are non-zero and we have both positive and negative eigenvalues, then the critical point is a saddle point with a *min-max* structure (see Figure 1 (b)). That is, if we restrict the function $f$ to the subspace spanned by the eigenvectors corresponding to positive (negative) eigenvalues, then the saddle point is a maximum (minimum) of this restriction.

- If the Hessian matrix is singular, then the *degenerate* critical point can be a saddle point, as it is, for example, for $\theta^3, \theta \in \mathbb{R}$ or for the monkey saddle (Figure 1 (a) and (c)). If it is a saddle, then, if we restrict $\theta$ to only change along the direction of singularity, the restricted function does not exhibit a minimum nor a maximum; it exhibits, to second order, a plateau. When moving from one side to other of the plateau, the eigenvalue corresponding to this picked direction generically changes sign, being exactly zero at the critical point. Note that an eigenvalue of zero can also indicate the presence of a gutter structure, a degenerate minimum, maximum or saddle, where a set of connected points are all minimum, maximum or saddle structures of the same shape and error. In Figure 1 (d) it is shaped as a circle. The error function looks like the bottom of a wine bottle, where all points along this circle are minimum of equal value.

A plateau is an almost flat region in some direction. This structure is given by having the eigenvalues (which describe the curvature) corresponding to the directions of the plateau be *close to 0*, but *not*

Figure 1: Illustrations of three different types of saddle points (a-c) plus a gutter structure (d). Note that for the gutter structure, any point from the circle $x^2 + y^2 = 1$ is a minimum. The shape of the function is that of the bottom of a bottle of wine. This means that the minimum is a "ring" instead of a single point. The Hessian is singular at any of these points. (c) shows a Monkey saddle where you have both a min-max structure as in (b) but also a 0 eigenvalue, which results, along some direction, in a shape similar to (a).

*exactly 0.* Or, additionally, by having a large discrepancy between the norm of the eigenvalues. This large difference would make the direction of "relative" small eigenvalues look like flat compared to the direction of large eigenvalues.

## 2 Reparametrization of the space around saddle-points

This reparametrization is given by taking a Taylor expansion of the function $f$ around the critical point. If we assume that the Hessian is not singular, then there is a neighbourhood around this critical point where this approximation is reliable and, since the first order derivatives vanish, the Taylor expansion is given by:

$$f(\theta^* + \Delta\theta) = f(\theta^*) + \frac{1}{2}(\Delta\theta)^\top \mathbf{H}\Delta\theta \tag{1}$$

Let us denote by $\mathbf{e}_1, \ldots, \mathbf{e}_{n_\theta}$ the eigenvectors of the Hessian $\mathbf{H}$ and by $\lambda_1, \ldots, \lambda_{n_\theta}$ the corresponding eigenvalues. We can now make a change of coordinates into the space span by these eigenvectors:

$$\Delta \mathbf{v} = \frac{1}{2} \begin{bmatrix} \mathbf{e}_1^\top \\ \dots \\ \mathbf{e}_{n_\theta}^\top \end{bmatrix} \Delta \theta \tag{2}$$

$$f(\theta^* + \Delta\theta) = f(\theta^*) + \frac{1}{2} \sum_{i=1}^{n_\theta} \lambda_i (\mathbf{e}_i^\top \Delta\theta)^2 = f(\theta^*) + \sum_{i=1}^{n_\theta} \lambda_i \Delta \mathbf{v}_i^2 \tag{3}$$

## 3  Empirical exploration of properties of critical points

To obtain the plot on MNIST we used the Newton method to discover nearby critical points along the path taken by the saddle-free Newton algorithm. We consider 20 different runs of the saddle-free algorithm, each using a different random seed. We then run 200 jobs. The first 100 jobs are looking for critical points near the value of the parameters obtained after some random number of epochs (between 0 and 20) of a randomly selected run (among the 20 different runs) of saddle-free Newton method. To this starting position uniform noise is added of small amplitude (the amplitude is randomly picked between the different values $\{10^{-1}, 10^{-2}, 10^{-3}, 10^{-4}\}$ The last 100 jobs look for critical points near uniformly sampled weights (the range of the weights is given by the unit cube). The task (dataset and model) is the same as the one used previously.

To obtain the plots on CIFAR, we have trained multiple 3-layer deep neural networks using SGD. The activation function of these networks is the tanh function. We saved the parameters of these networks for each epoch. We trained 100 networks with different parameter initializations between 10 and 300 epochs (chosen randomly). The networks were then trained using the Newton method to find a nearby critical point. This allows us to find many different critical points along the learning trajectories of the networks.

## 4  Proof of Lemma 1

**Lemma 1.** *Let $\mathbf{A}$ be a nonsingular square matrix in $\mathbb{R}^n \times \mathbb{R}^n$, and $\mathbf{x} \in \mathbb{R}^n$ be some vector. Then it holds that $|\mathbf{x}^\top \mathbf{A} \mathbf{x}| \leq \mathbf{x}^\top |\mathbf{A}| \mathbf{x}$, where $|\mathbf{A}|$ is the matrix obtained by taking the absolute value of each of the eigenvalues of $\mathbf{A}$.*

*Proof.* Let $\mathbf{e}_1, \dots \mathbf{e}_n$ be the different eigenvectors of $\mathbf{A}$ and $\lambda_1, \dots \lambda_n$ the corresponding eigenvalues. We now re-write the identity by expressing the vector $\mathbf{x}$ in terms of these eigenvalues:

$$|\mathbf{x}^\top \mathbf{A} \mathbf{x}| = \left| \sum_i (\mathbf{x}^\top \mathbf{e}_i) \mathbf{e}_i^\top \mathbf{A} \mathbf{x} \right| = \left| \sum_i (\mathbf{x}^\top \mathbf{e}_i) \lambda_i (\mathbf{e}_i^\top \mathbf{x}) \right| = \left| \sum_i \lambda_i (\mathbf{x}^\top \mathbf{e}_i)^2 \right|$$

We can now use the triangle inequality $|\sum_i x_i| \leq \sum_i |x_i|$ and get that

$$|\mathbf{x}^\top \mathbf{A} \mathbf{x}| \leq \sum_i |(\mathbf{x}^\top \mathbf{e}_i)^2 \lambda_i| = \sum_i (\mathbf{x}^\top \mathbf{e}_i) |\lambda_i| (\mathbf{e}_i^\top \mathbf{x}) = \mathbf{x}^\top |\mathbf{A}| \mathbf{x}$$

$\square$

## 5  Implementation details for approximate saddle-free Newton

The Krylov subspace is obtained through a slightly modified Lanczos process (see Algorithm 1). The initial vector of the algorithm is the gradient of the model. As noted by Vinyals and Povey (2012), we found it was useful to include the previous search direction as the last vector of the subspace.

As described in the main paper, we have $\frac{\partial \hat{f}}{\partial \alpha} = \mathbf{V}\left(\frac{\partial f}{\partial \theta}\right)^{\top}$ and $\frac{\partial^2 \hat{f}}{\partial \alpha^2} = \mathbf{V}\left(\frac{\partial^2 f}{\partial \theta^2}\right)\mathbf{V}^{\top}$. Note that the calculation of the Hessian in the subspace can be greatly sped up by memorizing the vectors $\mathbf{V}_i \frac{\partial^2 f}{\partial \theta^2}$ during the Lanczos process. Once memorized, the Hessian is simply the product of the two matrices $\mathbf{V}$ and $\mathbf{V}_i \frac{\partial^2 f}{\partial \theta^2}$.

We have found that it is beneficial to perform multiple optimization steps within the subspace. We do not recompute the Hessian for these steps under the assumption that the Hessian will not change much.

---

**Algorithm 1** Obtaining the Lanczos vectors

---

**Require:** $\mathbf{g} \leftarrow -\frac{\partial f}{\partial \theta}$
**Require:** $\Delta\theta$ (The past weight update)
   $\mathbf{V}_0 \leftarrow 0$
   $\mathbf{V}_1 \leftarrow \frac{\mathbf{g}}{\|\mathbf{g}\|}$
   $\beta_1 \leftarrow 0$
   **for** $i = 1 \rightarrow k-1$ **do**
      $\mathbf{w}_i \leftarrow \mathbf{V}_i \frac{\partial^2 f}{\partial \theta^2}$
      **if** $i = k-1$ **then**
         $\mathbf{w}_i \leftarrow \Delta\theta$
      **end if**
      $\alpha_i \leftarrow \mathbf{w}_i \mathbf{V}_i$
      $\mathbf{w}_i \leftarrow \mathbf{w}_i - \alpha_i \mathbf{V}_i - \beta_i \mathbf{V}_{i-1}$
      $\beta_{i+1} \leftarrow \|\mathbf{w}_i\|$
      $\mathbf{V}_{i+1} \leftarrow \frac{\mathbf{w}}{\|\mathbf{w}_i\|}$
   **end for**

---

## 6 Experiments

### 6.1 Existence of Saddle Points in Neural Networks

For feedforward networks using SGD, we choose the following hyperparameters using the random search strategy (Bergstra and Bengio, 2012):

- Learning rate
- Size of minibatch
- Momentum coefficient

For random search, we draw 80 samples and pick the best one.

For both the Newton and saddle-free Newton methods, the damping coefficient is chosen at each update, to maximize the improvement, among $\left\{10^0, 10^{-1}, 10^{-2}, 10^{-3}, 10^{-4}, 10^{-5}\right\}$.

### 6.2 Effectiveness of saddle-free Newton Method in Deep Neural Networks

The deep auto-encoder was first trained using the protocol used by Sutskever *et al.* (2013). In these experiments we use classical momentum.

### 6.3 Recurrent Neural Networks: Hard Optimization Problem

We initialized the recurrent weights of RNN to be orthogonal as suggested by Saxe *et al.* (2014). The number of hidden units of RNN is fixed to 120. For recurrent neural networks using SGD, we choose the following hyperparameters using the random search strategy:

- Learning rate
- Threshold for clipping the gradient (Pascanu *et al.*, 2013)

- Momentum coefficient

For random search, we draw 64 samples and pick the best one. Just like in the experiment using feedforward neural networks, the damping coefficient of both the Newton and saddle-free Newton methods was chosen at each update, to maximize the improvement.

We clip the gradient and saddle-free update step if it exceeds certain threshold as suggested by Pascanu *et al.* (2013).

Since it is costly to compute the exact Hessian for RNN's, we used the eigenvalues of the Hessian in the Krylov subspace to plot the distribution of eigenvalues for Hessian matrix in Fig. 4 (d).