[Reviews · NeurIPS 2014]

Submitted by Assigned_Reviewer_13

---------------------
Basically, I think it's really important to show that |H| is better than the Fisher/Natural Gradient matrix in an "apples to apples" comparison. Same setup, same learning rate heuristics, only different curvature. It's important because the Fisher matrix is known to be much better for nonlinear optimization than the Hessian. The experiment that you mention show that |H| is also much better than the Hessian. So the big question is, which is better: |H| or F, where both matrices are used in the same setup.

---------------------
Optimization methods assume that they are given convex approximations to the objective function. However, this assumption is clearly false due to the presence of the large number of saddle points in high dimensional spaces. The idea of explicitly modelling and dealing with saddle points is very interesting, and the results are intriguing: working with the absolute value of the eigenvalues really does seem to result in better performance on the tasks considered.

However, I feel that the experiments do not quite prove the point of the paper as well as it could have. Specifically, the experiments do not show convincingly that working with the absolute value of the Hessian is better than the Natural Gradient, or the Gauss Newton. To be precise, I noticed that Saddle Free Newton was initialized by SGD. In contrast, previous work on second order methods, including Martens (2010), did not initialize from SGD. And while you report better results, it is possible that most of the improvement was caused by the use of SGD as a “pretraining” stage of the second order. Thus the paper would be much more convincing if alongside SFN it included the same experiment (with the same optimizer, Krylov Subspace Descent), with the Gauss-Newton matrix. At present, there is only a comparison with damped newton.

The second issue, which is small yet significant, is that too much space and emphasis was placed on unnecessary formal justification of the method, introducing a new trust region method (which isn’t new --- the classical derivation of natural gradients minimizes the linear gradient over a quadratic constraint, which is precisely equivalent to an elliptical trust region method). Likewise, the |H| matrix has positive eigenvalues, and therefore dividing by this positive definite matrix is a proper optimization procedure.

I strongly recommend the authors to replace the Hessian with the Gauss-Newton / Natural Gradient matrix while keeping everything else fixed (and spending the appropriate amount of time tuning the new setting), in order to see whether it is the Hessian that contributes to the better performance, or whether it is the SGD pre-training.

Summary: The paper proposes an interesting and a new way of optimizing high dimensional objectives by explicitly handling saddle points and negative curvature. The experiments are interesting, and the results are good. However, the experiments do not prove the main point of the paper as well as it should.

Submitted by Assigned_Reviewer_43

Summary:

This paper argues that saddle points, and not local minima, are the primary difficulty in gradient- or Netwon’s method-based optimization of high dimensional non convex functions such as deep neural networks. The paper reviews related theoretical results, empirically demonstrates the problem in deep neural networks doing visual object recognition, and proposes a solution which is shown to be effective for training deep neural networks.

Main comments:

This is an original paper that could significantly reshape intuitions about learning in deep networks (and about other non convex optimization problems). The review of relevant theoretical work focuses on results that are not widely known in the deep learning community, and a useful contribution of the paper is the empirical verification on MNIST and CIFAR of the relationship between training error and index, showing that the result was not dependent on the simplifying assumptions used in the theory.

Is there a justification for only using the k biggest eigenvectors of the Hessian? It seems like the Krylov method is going to ignore directions with low or negative curvature that could be seen near saddle points. In particular, it seems possible that the Hessian could look fully positive definite in the krylov subspace when in fact there are other descent directions with negative eigenvalues. Consider the classical saddle from Fig 2a. My understanding is, if the Krylov subspace had dimension 1 it would point in the high positive curvature parabola even though the negative curvature direction exists. Hence the specific approximate SFN method proposed here seems like it could also be trapped by saddle points for which there are more positive, large eigenvalues than the Krylov subspace dimension.

It could be interesting to visualize the weights at the end of the MSGD epoch compared to a few iterations after SFN takes over. Is there a clear pattern to the changes, such as an unused unit with very small norm weights that becomes active? There may not be but if there is, it would be interesting to gain intuition: In practice is this mainly overcoming scaling symmetries across layers? Permutation symmetries among hidden units? Etc.

The paper does not particularly emphasize its result of lower error for the deep auto encoder compared to Hessian free optimization, but it should be noted that the Hessian free result did not use carefully tuned MSGD to initialize its behavior. This could have impacted the comparison.

The paper is clear and well written.

Minor:

It would be helpful to put a vertical line on Fig 4b where the switch to SFN occurs.
Summary: The paper convincingly argues that saddle points, not local minima, represent a key difficulty for current learning algorithms applied to non-convex models like deep neural networks. The paper highlights an algorithm able to more rapidly escape saddle points.

Submitted by Assigned_Reviewer_44

This paper argues about the existence of saddle points, which becomes the main burden that first order methods (such as SGD), and some second order methods (such as methods derived from Newton method) fail to exploit.

In particular, if a saddle point is encountered, an algorithm should exploit the direction of negative curvature to make progress: SGD methods will simply fail as they don't make much progress on directions of low curvature (as it has been argued in pervious art), while second order methods based on trust region will generally not exploit directions of (large) negative curvature, along which rapid progress could be achieved.

The authors propose a method that follows Vinyals and Povey's work, except that instead of using the Gauss Newton matrix, and regular Newton method on the Krylov subspace, they instead propose a new trust region method that generalizes to non-PSD Hessian matrices. They use the absolute value of the eigenvalues to set the trust region. This copes with saddle points in a much better way than SGD does (since it uses curvature), and much better than Newton based methods (since it can escape saddles naturally).

Some comments to the authors:
-At around line 315, perhaps clarify what you mean by k biggest eigenvectors - are they the biggest in eigenvalue absolute value? If not, how can the Krylov approximation capture negative curvature to be exploited by your algorithm?
-The improvement with SFN seems dramatic in, e.g., PTB. Could the authors report perplexities or some more standard metric on the held out set, for comparison's sake (carefully tuned SGD does very well on the PTB).
-Implementing these methods can be quite tricky. Are the authors planning on releasing code or a precise set of instructions with all the heuristics used?
-From a practical stand-point, how do things like regularization (e.g. dropout) affect saddle points?
-What was the network architecture, and how would the authors deal with ReLu nets? Can we say something about those in terms of second order behavior?
Summary: Good paper, tackling an important and general problem in machine learning and deep neural nets - optimization. This paper argues about saddle points being a major cause for first order methods (e.g. SGD) to suffer for slowness and stuckness. Enough evidence and fairly in depth analysis is given on several machine learning tasks that supports the main argument of the paper.
Author Feedback
Author rebuttal: Dear reviewers,

We would like to thank you for the thorough and insightful comments.

= R13 =

Q. "it is possible that most of the improvement was caused by the use of SGD as a 'pretraining' stage of the second order"
A. We believe that our experiments show SGD getting "stuck" around a saddle point, while the proposed saddle-free is able to achieve lower errors. The experiments say: "if we keep running SGD we do not see an improvement, while if we instead start our technique the error goes down". As well, in our experiments with RNNs "pretraining" with SGD and then training with damped Newton instead of SFN reduces the improvement by half. We will add these results to the paper.

Q. "there is only a comparison with damped newton."
A. That is true, and we intend to keep working on this approach to get more empirical evidence and improve our understanding of the issues sorrounding saddle points. This approach is based on a very different view on the optimization of neural networks (or even non-convex functions), one that might have some impact. We believe that the empirical evidence provided in this paper is enough to show that this new direction is worth exploring.

= R43 =

Q. "It seems like the Krylov method is going to ignore directions with low or negative curvature that could be seen near saddle points. .. method proposed here seems like it could also be trapped by saddle points for which there are more positive, large eigenvalues than the Krylov subspace dimension."
A. This might be true, and we do not believe that the Krylov approach we suggested is the final solution. Empirically, we observed that the Krylov subspace with a decent number of dimensions captured the directions with
negative eigenvalues as well.

Q. "Is there a clear pattern to the changes, such as an unused unit with very small norm weights that becomes active?"
A. Thanks for your insightful suggestion. Unfortunately we have not investigated such properties of the proposed algorithm. We strongly agree that it is worth looking at these properties in order to understand better the structure/properties of saddle points. We intend to do so in the future.

= R44 =

Q. "how can the Krylov approximation capture negative curvature to be exploited by your algorithm?"
A. This Krylov method captures the largest (by norm) eigenvalues of the Hessian. Provided that the norm of the negative eigenvalues are large enough they will be captured in this subspace. We empirically observed this behavior in the preliminary experiments.

Q. "Could the authors report perplexities or some more standard metric on the held out set"
A. We will add this information in the final paper. Due to memory constraints, we were only able to train small models, which resulted in a sub-optimal performance compared to the state-of-the-art performance (on the held out set) on this task.

Q. "Are the authors planning on releasing code or a precise set of instructions with all the heuristics used?"
A. We are planning to release an easy to use version of the code.

Q. "how do things like regularization (e.g. dropout) affect saddle points?"
A. That is an interesting question, for which we do not have a definitive answer at the moment. Understanding this will be important in the future to better understand the proposed learning algorithm as well as learning in general for deep neural networks.

Q. "how would the authors deal with ReLu nets?"
A. We can take the same approach as taken by SGD and ignore the discontinuity. ReLu units imply a very specific structure of the error surface, and we believe one may be able to take this structure into account to design a more efficient algorithm.